# Sex, Gender, and the Regulation of Prescription Drugs: Omissions and Opportunities

**DOI:** 10.3390/ijerph20042962

**Published:** 2023-02-08

**Authors:** Lorraine Greaves, Andreea C. Brabete, Mira Maximos, Ella Huber, Alice Li, Mê-Linh Lê, Sherif Eltonsy, Madeline Boscoe

**Affiliations:** 1Centre of Excellence for Women’s Health, Vancouver, BC V6H 3N1, Canada; 2School of Population and Public Health, Faculty of Medicine, University of British Columbia, Vancouver, BC V6T 1Z3, Canada; 3Women’s College Hospital, Toronto, ON M5S 1B2, Canada; 4School of Pharmacy, University of Waterloo, Kitchener, ON N2G 1C5, Canada; 5Neil John Maclean Health Sciences Library, University of Manitoba, Winnipeg, MB R3M 3M1, Canada; 6College of Pharmacy, Rady Faculty of Health Sciences, University of Manitoba, Winnipeg, MB R3M 3M1, Canada; 7Cochrane Sex/Gender Methods Group, Ottawa, ON K0A K4C, Canada; 8Women and Health Protection, Ottawa, ON K0A K4C, Canada

**Keywords:** sex, gender, equity, SGBA+, regulation of prescription drugs, pharmacovigilance, ICH, FDA, EMA, Health Canada

## Abstract

The regulation of prescription drugs is an important health, safety, and equity issue. However, regulatory processes do not always consider evidence on sex, gender, and factors such as age and race, omissions that advocates have highlighted for several decades. Assessing the impact of sex-related factors is critical to ensuring drug safety and efficacy for females and males, and for informing clinical product monographs and consumer information. Gender-related factors affect prescribing, access to drugs, needs and desires for specific prescribed therapies. This article draws on a policy-research partnership project that examined the lifecycle management of prescription drugs in Canada using a sex and gender-based analysis plus (SGBA+) lens. In the same time period, Health Canada created a Scientific Advisory Committee on Health Products for Women, in part to examine drug regulation. We report on grey literature and selected regulatory documents to illustrate the extent to which sex and gender-based analysis plus (SGBA+) is utilized in regulation and policy. We identify omissions in the management of prescription drugs, and name opportunities for improvements by integrating SGBA+ into drug sponsor applications, clinical trials development, and pharmacovigilance. We report on recent efforts to incorporate sex disaggregated data and recommend ways that the management of prescription drugs can benefit from more integration of sex, gender, and equity.

## 1. Introduction

Prescription drugs are highly regulated in most countries using systematic regulatory processes to assess applications for drug approval from industry or research sponsors. Regulators assess preclinical and clinical trials data in such submissions, and post approval, monitor impact of drugs through pharmacovigilance and adverse event (AE) reporting. Approvals by agencies such as the Food and Drug Administration (FDA), Health Canada (HC), and the European Medicines Agency (EMA) can be prolonged and detailed, in order to assure safety and efficacy for those consuming drugs. Since 1990, the International Council on Harmonization (ICH) has brought together regulatory authorities and pharmaceutical industry observers and member to standardize some regulatory processes and share knowledge.

Sex and gender related factors are important to understanding the safety and efficacy of prescription drugs for both sexes and all genders. Historically, several groups have been excluded or underrepresented in clinical trials, including women, pregnant, lactating, pediatric and geriatric populations, and racial/ethnic minority groups. While these exclusions are slowly being recognized and remedied, there remain significant questions about how currently prescribed drugs affect the entirety of the population directly resulting from the lack of representativeness of clinical trials, and inadequate pharmacovigilance.

There are other considerations in the management of drugs. Sex-related factors affect pharmacokinetics and pharmacodynamics (PK/PD) processes as well as the occurrence and reporting of adverse drug reactions (ADRs) and events (AEs) [1,2]. Gender related factors influence marketing and may affect desire for, and the prescribing of drugs as well as the reporting of AEs. However, integrating such evidence and analysis throughout the lifecycle management of prescription drugs is not standard practice.

Advocates for including sex and gender in the regulation of drugs and women in clinical trials have based their demands on sex-specific examples of missing evidence, AE reporting, catastrophic drug events particular to women, and the need to adhere to SGBA+ of policy, practice, and research. Over the past 50 years, feminist activist women’s health groups have consistently argued for expanding the parameters of prescription drug research, clinical trials, and ultimately regulatory processes to include females and women, as well as pregnant and lactating women [3,4]. Subsequently, calls for more representative research including minority groups [5], pediatric [6] and geriatric populations have been made [7]. Countries such as Canada have invoked SGBA+ policies, requiring that federal processes and policies be assessed prior to and during implementation for their differential effects and impacts on women, men, boys, and girls [8]. Similarly, research funders in several countries have set requirements for integration of sex and gender related variables in health research [9,10], and journals have set requirements for reporting on these issues [11].

All of these changes reflect advancements in sex and gender science, and increased knowledge about sex differences and sex-related factors affecting prescription drug use, highlighting research gaps and issues that often disproportionately and negatively affect women. In brief, sex and gender science is a growing specialty that encompasses several research approaches, including identifying and measuring sex and gender differences, sex-related factors, sex/gender interactions, sexual and gender minority health, and intersectional approaches that centre sex/gender [12]. Various policy initiatives are critical to addressing health and gender equity, including SGBA+, equity, diversity, and inclusion (EDI) initiatives, and equity plans and frameworks, especially those with gender transformative goals [13]. SGBA+ is an analytic process that, when applied to health, involves assessing programs, policies, treatments, prevention, messaging, and health promotion for differential impacts on people based on sex, gender, and a range of diversity factors, with a view to tailoring such efforts to achieve more equity [12].

In this context the Canadian government created the Scientific Advisory Committee on Women’s Health Products (SAC-HPW) in 2019 to advise Health Canada on changes in regulatory systems [14]. The SAC-HPW is comprised of appointed clinicians, scientists and advocates and has worked with Health Canada representatives to integrate more SGBA+ into regulatory processes and practices, to advise on representation and inclusion in trials, improved pharmacovigilance and safety of drugs and devices for women, and enhanced communication with clinicians and consumers. In the past 3 years, the SAC-HPW has made recommendations for more reporting and analysis of sex and gender related factors in the management process, enhancements to a SGBA+ action plan, and on building capacity within regulatory personnel [15].

Several discussions of the importance of sex and gender related factors, age and race have occurred in this context. These have included how sex impacts prescribed drugs, as a biological variable with several components that relate to assessing processes or factors affecting either differential or mechanistic impacts of prescribed drugs [16] such as hormones, enzymes, genetics, neurobiology, anatomy, and physiological processes. This approach goes well beyond using sex as a demographic signifier in clinical or pharmacovigilance databases and considers more nuanced sex-related factors and processes that directly impact the safety and efficacy of prescribed drugs on bodies and people. Similarly, gender has been discussed as a multi-faceted social variable that has a direct impact on understanding access to, use, prescribing and marketing of prescription drugs. Gender encompasses gender relations, roles, norms, and identities, along with how gender is embedded in institutional practices such as laws, education and media [16].

## 2. Materials and Methods

This article draws on a partnership project on sex, gender and the lifecycle management of drugs aimed at evaluating how sex and gender are included in regulatory processes, including an analysis of adverse event reporting and postmarket vigilance that has been reported elsewhere [17]. The project framework was the Canadian SGBA+ Health Policy-Research Partnerships, funded by the Canadian Institutes of Health Research and Health Canada, designed to bridge the gaps between research knowledge and policy development and support the rigorous application of SGBA+ to ensure Health Canada’s activities address the diverse needs of women, men, girls, boys and gender-diverse people, to improve the health of all Canadians.

We undertook an overarching scoping review to identify relevant research on sex, gender and the lifecycle management of drugs [17]. A systematic search was performed in the following databases: MEDLINE (Ovid), Embase (Ovid), Cochrane Library (Wiley), International Pharmaceutical Abstracts (Ovid), CINAHL (EBSCO), and Scopus to identify relevant studies published in English between 2010 and 2020 (up to 23 July 2020). A combination of keywords and Medical Subject Headings (MeSH) terms on Sex and Gender (including gender identity, sex factors, and sex characteristics) and the Drug Lifecyle (including research and development, discovery, design, legislation, costs, and industry) was used. The detailed search strategy has been published elsewhere [17].

The searches resulted in *n* = 8508 unique records. First, we screened them by title and abstract and then the full texts of the papers were retrieved, and full papers screened for inclusion. The title and abstract screening was conducted by a single reviewer, and full texts were screened independently by two reviewers. The team worked on a set of inclusion criteria that were amended based on increasing familiarity with the literature and the iterative screening process we undertook to select relevant studies. The inclusion and exclusion criteria have been published elsewhere [17].

In total, *n* = 98 papers on processes on lifecycle management of drugs were identified in the academic literature search and mapped to each phase of the lifecycle management of drugs: clinical trial phase, submission review, monitoring and intervention and pharmacovigilance. Of the 98 included papers, *n* = 63 were on PK/PD mechanisms related to the clinical trial phase and monitoring and intervention and *n* = 35 papers were on adverse drug reactions and/or adverse events.

We also conducted targeted searches of grey literature and relevant organizational website. Based on our own and our policy partners’ knowledge, two relevant websites were identified. Thirty-one documents from four regulatory agencies were selected with our policy partners as directly relevant to our project. In this paper, we provide a critical analysis of all of this content using a SGBA+ [8], drawing examples from the following:Eleven articles identified in the scoping review that illustrate SGBA+ issues in the evidence regarding PK/PD and ADRs/AEsThirty-one publicly available documents from Health Canada, ICH, FDA and EMAFour documents from two websitesNineteen editorials, commentaries, and letters from the scoping review [17]


## 3. Results

In many of the academic papers, the terms referring to sex and gender were either misused, conflated, or exchanged, contributing to poorer quality analyses, and reporting and interpretation challenges and potentially diminished replicability. These issues are not unique to the literature on prescription drug regulation, and are indeed endemic to much health research, analysis, and reporting. While such practices are gradually being diminished by initiatives from research funders requiring sex and gender inclusion and offering training [18], legislation requiring inclusion in clinical trials [9], recommendations about measuring sex, sexual orientation and gender (identity) [19], and journals requiring full reporting and analysis of sex and gender [11], there are still many instances of inadequate or imprecise data collection, measurement, concept and language misuse, and partial reporting.

### 3.1. Pharmacokinetics and Pharmacodynamics (PK/PD) of Drugs

These conceptual issues were present in the PK/PD literature surrounding prescription drugs. For example, the aim of one UK study was to probe “gender-related” differences in PK of doxorubicin in young people [20]. Another study conducted in the US examined the impact of age, gender, and weight (covariates) on the PK of 3-aminopyridine-2-carboxyaldehyde thiosemicarbazone, an anticancer drug [21]. While both studies found sex affected the pharmacokinetic profiles of the drugs, they described such impacts as “gender”.

Further, the main paradigm underpinning research in PK/PD studies and sex/gender was one of ‘sex differences’ (comparing male and female) despite the rich area of potential investigation regarding sex-related factors that PK/PD processes represent and that are embedded in sex as a biological variable (SABV). For example, a US study aimed to examine the PK in males and females of an antifungal agent [22]. In that study, there were no differences in drug clearance and exposure between nonelderly males, elderly males, and nonelderly females. On the contrary, findings showed that elderly females had the highest exposure of all groups tested, and the highest number of reported treatment emergent AEs [22]. While such sex differences studies are important signals, they need to be followed up by studies to better understand the mechanism(s) of drug variability by sex and the potential clinical implications.

We also found studies conducted with males (to avoid female menstrual cycle variations) that extrapolated results to females For example, a US study reported that “female subjects were excluded to eliminate any heterogeneity that could have arisen secondary to menstrual cycles and hormonal variations” [23] but concluded that: “demonstration of PK and PD equivalence in the most sensitive population of healthy male subjects supports that equivalent epoetin concentration profiles and reticulocyte response profiles can be expected after equivalent doses of Epoetin Hospira or Epogen in treated male or female patients” [23]. There are no details provided by the authors regarding what “sensitive” criteria were applied in order to support extrapolation of data to the general population.

Issues such as these permeated the literature we reviewed, often requiring us to reinterpret the language used to match it to the study parameters or indicators, and report accordingly. These examples of misuse and conflation of sex and gender, and exclusions of females, lead to weaknesses in generalization and undermine the utility of research results. These omissions and errors spill over into the measurement of AEs, along with fundamental weaknesses in pharmacovigilance systems, reporting practices, and measurement practices.

### 3.2. Adverse Drug Reactions (ADRs) and Adverse Events (AEs)

Sex and gender are not systematically included in the pharmacovigilance systems, and data collection and reporting of ADRs and AEs differ from country to country and in each pharmacovigilance database. These systems do not contain exhaustive counts of actual AEs, as actual numbers of AEs in a given country or regarding a given drug are typically unknown. Some countries regulate mandatory reporting from health care providers and health care institutions, and some mount public awareness campaigns to encourage the public to voluntarily report. Some require information on sex and gender, verify or validate reports, or remove duplicates, and others do not. In the evidence we found regarding AEs, there were very different systems forming the basis of research studies or reports [17,24].

Nevertheless, assessing AE reports by sex and gender (and often also age and race) is essential in order to identify or verify patterns of reactions, in particular those linked to sex related PK/PD. But it is made difficult by the nature and limitations of reporting systems. For example, the rate of prescribing a particular drug or drug class by condition and sex and/or gender in a specific country is not usually clear, nor is the actual overall denominator of prescriptions for a particular drug typically made available. Hence, there are many caveats in interpreting AE data, and even more in comparing between countries.

We found some evidence on AEs that mention sex or gender or other equity characteristics by drug classes or drugs. Reports of AEs/ADRs across different countries show that there are more reports concerning women than men, although with some drugs men are more likely to present serious AEs/ADRs. For example, females reported more cases of AEs associated with anticancer drugs than males, but data reveal more cases of serious ADRs among males [25]. Some drug classes were reported as causing more AEs among women. For example, women had a higher prevalence of ADR reports in 6 of the 10 groups of antihypertensive drugs [26]. Sex/gender differences have also been observed in the likelihood of reporting an AE or ADR after receiving a vaccine, the seriousness of the report, the type of AE/ADRs reported, and the reported symptoms [27].

When intersections of sex and other variables are analyzed, there are different patterns for women and men across age categories. For example, both sex and age impact the differences found in the ADRs from vaccines and other drugs [28]. ADR reports from vaccines and other drugs were reported most commonly by females aged 10 to 64 and males from older age groups [28]. Although there is evidence that sex is highly relevant in ADR reports, variables such as sex (and age and race/ethnicity) are inconsistently included in pharmacovigilance databases affecting the interpretation of the results [29,30].

The structure and regulation surrounding pharmacovigilance systems is critically important to understanding the differential impacts of drugs by sex and gender postmarketing. However, despite the need for such information by clinicians and consumers, there is considerable room for improvement in designing AE reporting systems and regulating reporting to provide more meaningful, validated, inclusive, reportable and comparable data [19].

### 3.3. Documents from Selected Regulatory Agencies

Building upon the evidence regarding the importance of how sex and gender related factors affect PK and PD processes, and AEs, we report on representative regulatory documents from four agencies (US (FDA), Health Canada (HC), the EMA (EU) and the ICH) and probe the treatment of sex, gender and other important factors regarding diversity and equity in these documents.

#### 3.3.1. International Council of Harmonization

The International Council of Harmonization of Technical Requirements for Pharmaceuticals Human Use (ICH), established in 1990, provides guidance documents and other policy statements to numerous participating countries. We observed several drawbacks with the ICH documents. For example, the ICH documents often use sex and gender interchangeably, with some exceptions. While the ICH documents focus on reproductive and fetal health issues when considering women, they do not integrate more comprehensive sex and gender related factors and intersectional variables such as ethnicity, or age [31,32]. More fundamentally, when considering sex related factors in clinical trials, the ICH does not explicitly grapple with the implications of “what we do not know” in sex and gender science when forming its conclusions.

For example, a key 2004 ICH document, revised in 2009 addresses sex-related considerations in clinical trials [33]. The revision distinguished between sex and gender, correcting the conflation of terms in the 2004 version, and refers to previous statements regarding the importance of addressing pediatric and geriatric populations, but recognized that the ICH had overlooked ‘women’s’ concerns. Although the ICH acknowledged that women were less likely than men to be included in phase I of clinical trials, and that it was difficult to determine the full extent of sex/gender related population burden of disease, it concluded that since many of their key safety and dosage guidelines mentioned sex/gender, there was no need to create sex/gender specific guidelines. This position indicates a lack of understanding of the extent of sex and gender-related factors affecting both PK/PD processes as well as a lack of understanding of gender and the role it might play in drug research [33].

A 2016 version of a 1995 ICH document on good clinical practices gives guidance on conducting clinical trials with humans, but lacks a focus on sex and gender [34]. A related 2019 document includes sections on engaging patients to provide meaningful input on real life experiences, engaging patient organizations, and on “special populations”, including pregnant women, nursing women, pediatric and geriatric populations, and those with renal or hepatic impairment, but again with no mention of sex and/or gender related factors affecting PK/PD or engagement issues [35].

ICH documents on pediatric efficacy issues [36] address sexual maturation, estrous variability, and sperm health, but no sex related factors in PK or PD are mentioned. The 1993 geriatric guidance included suggestions regarding the assessment of PK/PD issues in patients [32], but again, no sex-related factors are mentioned. The lack of SGBA+ in the geriatric guidance indicates a limited understanding, or application of the impacts of sex-related and age-related factors over the lifecycle [32].

The ICH issued guidance on trial safety in 2010 [31], suggesting safety reporting during clinical trials be filed annually by the industry sponsor(s) and data collection by sex, age and race. However, it misuses the term gender [31]. Another ICH document describes principles related to clinical trial operation [37]. It states that: “*principles delineated in this guidance deal with minimising bias and maximising precision*” but does not achieve this with respect to sex and gender and does not integrate sex and gender science, SGBA+, or any intersectional variables such as age or race. It does not include statistical approaches that investigate sex-related factors as discovery variables, but only as they may or may not affect the dominant variable. This document does suggest subgroup analyses with respect to age and sex, but cautions: “*In most cases … subgroup or interaction analyses are exploratory and should be clearly identified as such; they should explore the uniformity of any treatment effects found overall*” [37]. An addendum to this document released in 2019 [38] focused on clarifying estimates of treatment effect but does not mention any issues or measurement principles related to sex and gender science, or SGBA+. Numerous safety guidelines have been produced by the ICH. One 2020 example provides guidance on assessing pre-clinical studies with respect to reproductive toxicity that differentiates between male and female animals and summarizes data on reproductive toxicity according to a range of measures that might impact both male and female fertility and reproductive processes [39].

ICH guidance on the consideration of ethnic factors in bridging studies was issued in 1998 [40] and addressed data issues and PK/PD factors that might be impacted by ethnicity and could lead to potential effects on drug use, dosage, safety, and efficacy. Ethnic factors may be important in situations where intrinsic factors such as genetics, age, gender (sex), height, weight are evident. Extrinsic factors such as diet, tobacco and alcohol use, compliance with medication regimes, or socioeconomic status (SES) are also included. While this document primarily addresses ethnicity, it does recognize that gender (meaning sex) interacts with ethnic-related factors as one avenue of impact and illustrates this as a genetic consideration. However, it does not explicitly integrate gender or sex considerations into its discussion of pharmacokinetics or pharmacodynamics [40].

The ICH also addresses post approval pharmacovigilance [40,41,42] and recommends that data related to gender (meaning sex), age, weight, and height should be collected, along with medication identifiers and concomitant treatments as ideal elements of reporting adverse events. However, this recommendation is not reflective of basic PK/PD or any relevant gendered factors [42]. A document advising sponsors on pharmacovigilance activities including the structure and design of observational studies mentions reproductive/developmental toxicity as a key reportable element. While this document conflates sex and gender it does indicate that epidemiological data collected by sex should be included, “whenever possible” [43].

#### 3.3.2. The Food and Drug Administration (USA)

While early FDA documents conflated the terms gender and sex, or used them interchangeably, since 2012 efforts have been made to clarify these terms in documents. Changes in legislation in 1993 [44] addressed historical exclusions of pregnant women in clinical trials and in the past 12 years, attention has been paid to improving the inclusion of minority groups (race/ethnicity categories), pediatric populations, and older people [45].

As early as 1985, FDA required submission of safety data “presented by gender, age, and racial subgroups” and descriptions of statistical analyses used to acquire such data [46]. Although the term gender was used to describe biological sex, the regulation explicitly required that data for both efficacy and safety be disaggregated by sex, age, and race/ethnicity. The sponsors were required to identify differences in dose or dose interval among subgroups. Succinct and explicit instructions for sex-disaggregated analysis left little room for alternate interpretations. The document frames this as a requirement and should prompt sponsors to investigate subgroups thoroughly prior to application [46]. A 2002 document on enrolment reporting required sex and subgroup data but no explicit requirements were made for ongoing adverse events reporting [47].

In a 2009 regulatory document on clinical trial holds (interrupting or delaying recruitment, enrolment, and administration of trial drugs), some of the requirements included the reporting of disaggregated data in cases where reproductive potential was an issue for females or males. In this document, the term gender was used instead of sex and single-sex studies were referenced. Studies that targeted only one sex were allowed for research on sex-related issues concerning one sex such as drug excretion in semen or effects on menstrual function or if a study included only one sex, but another study with the other sex had been completed or was running concurrently. However, no standards for pediatric or geriatric populations existed [48].

In 2012, the Food and Drug Safety and Innovation Act established rules requiring the reporting of monitoring sex-disaggregated data. It also referred to other subgroup disaggregation stipulating that the information must be disseminated and accessible to a range of audiences. The Act specifically required the FDA to submit a report to Congress on how well drug companies disclosed data based on sex, age, and ethnicity. It also required the FDA to make these data available to health care providers, researchers, and patients. It also monitored how subgroup analyses based on sex, age, race, and ethnicity were presented in new drug applications, and if data on product safety and effectiveness by subgroups (sex, age, race/ethnicity) are made available to the public in a timely manner [49].

In 2014, an action plan was published to improve completeness and quality of demographic subgroup data collection, reporting, and analysis. The plan’s goals included identifying barriers to subgroup enrollment in clinical trials and strategies to encourage participation; and increase transparency of demographic subgroup data [50]. In an effort to improve trial recruitment and representativity, in 2020, the FDA issued an important guidance document [51] which aimed to broaden eligibility criteria while maintaining safety and efficacy standards. In a footnote, the FDA addressed prior conflation of sex and gender terms, stated the difference between sex and gender, and named sex and gender as separate clinically relevant subgroups [51].

#### 3.3.3. European Medicines Agency (EMA)

In 2014, the EMA issued regulations for the European Union (EU) [52] to ensure enhanced harmonization of rules for conducting clinical trials throughout the EU. These regulations instituted a streamlined procedure for trial application submission, assessment of the trial, protection of participants, informed consent, and requirements for transparency. In this document, the term gender is used in place of biological sex, but there are requirements for participation of both sexes, representative of the population who would use the product. The protocol requires disclosure of subgroups (including age and sex), a justification of sex and age allocations and any exclusions, and clinical trial data must be disaggregated by sex and age. There is no mention of race/ethnicity or socioeconomic status [52].

In 2019, the EMA published different principles and assessment strategies for the inclusion of subgroups recommended to use subgroup analyses to estimate or test a treatment effect and compare them across subgroups [53], but again, the term gender is used in place of biological sex. While there are acknowledgements that subgroup analyses will be made at minimum based on sex, age, and ethnic origin, there are no stated SGBA+ considerations [53].

#### 3.3.4. Health Canada

Clinical trial guidance pertaining to inclusion of women and sex differences was first published in 1997, revised in 2013 [54], and is under review at time of writing. These guidelines are in place for the industry and researchers. They point out the importance of a balanced participation in clinical trials and reflect growing evidence regarding ‘sex differences’ in PK and related sex and gender science impacts. It also responds to the ongoing exclusions or limited numbers of women in trials and encourages applicants to document sex differences in therapeutic results, to include both males and females, and to enact single sex trials when warranted. Based upon a ‘sex difference’ paradigm, the guidelines delineate specific steps for understanding and interpreting trial data including that null sex differences do not necessarily mean there are none. The guidelines mention, among other things, the need for information on drug impacts in pregnant and breastfeeding women and offers detailed approaches and safeguards for increasing inclusions. However, little attention is given to male fertility and reproductive potential, or male sperm contribution to fetal and infant/child health [54].

The 2013 guidance outlined the responsibilities of Trial Sponsors of Clinical Trials Applications [55]. This document focused on operational issues of clinical trials, but did not mention sex or gender, SGBA+, or propose that sex and/or gender related factors (or intersectional factors such as SES or race, for example) may impact clinical trial design and operation. The procedural guidance for submission omitted any reference to terms such as sex, gender, males, females or women or men, or SGBA+ requirements for reporting or filing management related applications and submissions [56] but did require that research ethics boards be composed of both “men and women”. However, the 2013 Guidance document on clinical trial inclusion [54] indicates the following considerations of relevance to submissions:


*“Where sponsors intend a therapeutic product to be used by both women and men, it is recommended that sponsors include both sexes in (a) nonclinical studies; and (b) in clinical trials to allow detection of potential sex-related differences in efficacy and in safety.”*


Even though product monographs are the main vehicle for providing information on the effects of a drug product to clinicians and consumers, the guidance regarding product monograph development was lacking any sex and gender related factor mentions or implications. While there is mention of both males and females with respect to toxicology and reproduction, there was no inclusion of sex-related adverse event reporting [57]. While PK/PD issues are the backbone of pharmaceutical assessments, there is no explicit sex-related information or guidance on these aspects in many of the documents and review forms. The guidance on clinical trials covers post-market monitoring along with several other documents [58]. It also requests data availability regarding gaps in knowledge on “sex based potential differences or considerations” [54]. Mandatory reporting of serious AEs and adverse drug reactions by hospitals, including patient’s age and sex, as distinguished from gender identity was instituted in 2019 (also known as “Vanessa’s law” in memory of a 15 year old woman who died of an arrythmia after taking cisapride as prescribed) [59]. However, a similar document regarding manufacturers’ reporting of adverse events does not address sex and gender or SGBA+ considerations in its mandate [60].

### 3.4. Grey Literature on Women and Health Protection and Multi-Regional Clinical Trials Center Websites

In this section, we include material relevant to the lifecycle management of drugs from two websites, both reflecting decades of equity-seeking advocacy in the area of management of prescription drugs and related issues.

#### 3.4.1. Women and Health Protection

The Canadian Women and Health Protection working group [61] was established in 1998 and active until 2010. The working group was part of the (then) federally funded Centres of Excellence for Women’s Health Program, and its aim was to produce numerous resources and reports, some of which are still highly relevant. Researchers, advocates and women’s health providers formed part of this working group. They functioned as a watchdog of pharmaceutical drug legislation in Canada and was particularly concerned with the many issues related to the lifecycle management of drugs, such as drug approval processes, inclusion of women in clinical trials, and post-marketing surveillance programs.

The website is a legacy of their work and contains resources that applied a women’s health lens and a gender-based analysis to a issues in health protection, including prescribed drugs. For example, the group played a significant role in bringing attention to violations of the direct-to-consumer advertisement of prescription drugs regulations in Canada and in holding the government accountable for enforcing the legislation [62]. They also analyzed the international harmonization of pharmaceuticals regulation and how it could impact health regulation in Canada and the safety, effectiveness, and availability of medications for women by neglecting to provide guidelines on the inclusion of women in clinical trials [63].

Ford et al. (2004) proposed several recommendations for improving the management of prescription drugs in Canada. First, they urged that all drugs be analyzed using a gender-based analysis (the term used in the Canadian 1999 *Women’s Health Strategy).* They also suggested that all reviewed data used in decision making should be published, and that a robust consumer adverse event reporting system should be established. Further, in 2005, they also suggested that the results of clinical trials should be made public [64], which various stakeholders continue to call for today. Due to a loss of core funding in 2010, the group is no longer active in observing the management of prescription drugs in Canada. Interestingly, in 2004, the group argued that the overall protection system was not inadequate because of the regulations themselves, but rather because the regulations in place were enforced and applied improperly [65].

#### 3.4.2. Multi-Regional Clinical Trials Center

The Multi-Regional Clinical Trials Center (MRCTC) of Brigham and Women’s Hospital and Harvard (https://mrctcenter.org) (accessed on 21 October 2021) was established in 2009 at a summit initiated by Pfizer Inc and now functioning in a multi-source funded academic environment, aimed at improving the operation of clinical trials. A key guidance document by Bierer et al. (2021) [66] clarifies the importance of diversity, inclusion, and equity in clinical research and suggests actions to achieve this goal. They argue that inclusion occurs disproportionate to the populations intended to use the drug, and that underrepresentation or lack of access affects the results of trials, noting that sex is a biological construct and essential to inclusion and race/ethnicity are social constructs that are often proxies for genetic, social or economic factors. The authors note a lack of diversity in clinical trials in terms of race, ethnicity, sex, gender, the elderly, the young, and genetics, and that investigation of these factors requires validated methods of categorization and proper language (i.e., using sex and gender terms).

They also note that it is still unknown as to whether sex differences affecting the safety and effectiveness of drugs are due to pharmacokinetic, pharmacodynamic, pharmacogenomic differences, hormonal differences, polypharmacy, or other factors. The authors suggest that as this research expands, a plan should be put in place to expand inclusion criteria accordingly [66].

Beyond inclusion, sufficient numbers of diverse participants are needed to perform subgroup analyses through pooling series of trials in early product development, or by leveraging pharmacogenomics research, meta-analyses, data repositories, and other data sources. They suggest that trial planning consider these factors in early stages, build on evidence of at-risk populations, and increase inclusion by identifying barriers, creating resources, approaches and technology to address barriers, evolving study designs (for example, to be sensitive to evidence of sex differences), and standardizing data terminology, collection, and analyses.

Actions are suggested such as recommending that reference values should be adjusted if there are known differences among subgroups, such as known sex differences in average weight and height, and that *each* eligibility criterion should require justification, be re-evaluated at each level of review, and that a searchable repository of average values for subpopulations of important variables (ex. average weight for men and women) be created. They also recommend meaningful engagement with underserved and underrepresented patients through partnerships with caregivers, patient advocacy groups, and community organizations contributing to the recruitment and retention of these populations, setting priorities for research, designing and conducting studies, and in dissemination [66].

In the guidance document, diversity and inclusion are deemed important for individual clinical trials, but also for each stage of the product lifecycle and for the workforce and organizations involved at each level. Different considerations at each stage of development are noted, ranging from biologic elements and mechanisms in early phases (presumably including sex), to conducting trials post-market to investigate biologically relevant diversity issues. Finally, they suggest integrating real world evidence to explore underrepresented populations in pharmacovigilance phases and to increase diversity in the workforce carrying out this work. Within the guidance document, the authors note that while US regulatory agencies provide guidance about diversity and inclusion, they do not mandate it, and recommend creating goals, requirements, workforce training, and metrics to measure progress. This guidance document provides practical ways to increase diversity and inclusion in the lifecycle management of drugs and can be applied through a sex and gender lens at all levels [66].

### 3.5. Grey Literature Identified in the Scoping Review

We also examined grey literature derived from the academic searches that referred to sex and gender-based factors and the lifecycle management of drugs. These grey articles include commentaries, letters, opinion pieces and policy comments, and are useful for understanding current critiques of the lifecycle management of drugs.

#### 3.5.1. Policies and Practices

Some articles described policy changes that were put into place and highlight the inclusion of women in clinical trials and sex-based analyses. However, more actions are needed to address sex differences in ADRs, the continued lack of inclusion of women in phase I clinical trials, the lack of inclusion of pregnant and lactating women [67], and a lack of research that supports precision medicine. Furthermore, there is also a need for more consideration of how gender related factors influence the lifecycle management of drugs, in addition to sex related factors. Despite the many articles in the grey literature that dealt specifically with SGBA+ concepts, many of them were flawed by persistent misuse of language.

As mentioned in the Women and Health Protection website, enforcement can be an issue. A 2010 commentary by Fisher and Ronald showed how regulatory agencies can be affected by gender biases leading to the perpetuation of inequalities if safety and evidence standards are not enforced, even when these standards are in place, the inclusion of women in clinical trials is recommended, and some version of (S)GBA+ is recommended as is the case of Canada and the United States [68]. Even though the FDA requires new drug applications to include information on sex differences, the authors suggest that the FDA has often failed to enforce this requirement leading to drugs being approved before there has been enough research on potential sex differences in drug safety and efficacy [68]. Furthermore, gender norms and politics can influence the enforcement of the standards. For example, the FDA prioritized the review of sildenafil citrate (Viagra) and approved it within 6 months but mifepristone, a safe and effective drug for inducing abortion, was repeatedly delayed as a result of gender biases regarding sexual behaviour and activity in women and men [68]. In 2012, the FDA prioritized the development of treatments for female hypoactive sexual desire disorder as part of their commitment to improve women’s health [69].

After exploring the commitment of the pharmaceutical companies and academic medical institutions in the US in diversifying their workforces and providing inclusive environments, Ahmed, Strauss, and Bierer (2020) published a letter to the editor in the Therapeutic Innovation & Regulatory Science journal to share their findings. They found that the top 20 pharmaceutical companies and the top 10 academic medical institutions made such statements public. However, when they searched for similar statements about diversifying clinical research populations to reflect those for whom treatment is intended, they were unsuccessful, noting that while public statements and commitments are a first step, setting standards, creating cultural shifts, and holding organizations accountable are critically important [70].

With increasing awareness and guidance on the importance of including diverse populations and representing women in clinical research, why do women still encounter barriers such as underrepresentation in some phases of trials? Some researchers have identified prevailing misconceptions about sample size requirements as a barrier to inclusion [71] whereas others have pointed to caretaking responsibilities as a barrier to recruitment and participation [68]. Even though the Canadian Institutes of Health Research requires grant applicants to include sex and gender in their applications and supplementary financial aid is provided for this purpose, many journals still do not include similar requirements that mirror these standards of reporting and data analysis. Nowogrodzki (2017) emphasized that ultimately journals and peer reviewers need to evaluate the inclusion of sex and gender in clinical studies as a way of holding researchers accountable and advancing these issues in clinical research [71].

Underrepresentation of women has been particularly rampant in phase I clinical trials, which has resulted in less efficacy and safety information for women compared to men and has led to an increased risk for ADRs [71]. This has important consequences since key decisions regarding safety and dosing are made during phase I clinical trials and if women are not adequately represented and outcomes are not sex-disaggregated, dosing is skewed to the male body [68].

The underrepresentation of pregnant and lactating women in most clinical trials is also reflected in the fact that such trials often require proof of a negative pregnancy test and the use of contraception as eligibility criteria, reducing access and underlining exclusion. Despite existing guidelines in Canada, the US, and Europe on ensuring that pregnant and breastfeeding women are included in clinical trials as well as a desire among pregnant women and their healthcare providers to be included in clinical trials, this exclusion contributes to a lack of safety and efficacy information available to physicians and women when making treatment decisions, and resulting untreated diseases have potential to harm both a woman and her fetus [67].

#### 3.5.2. Pharmacokinetics and Pharmacodynamics (PK/PD)

Some articles in the grey literature pointed to the importance of investigating sex differences in the PK/PD of drugs to advance precision medicine. A commentary published in 2017 discussed sex differences in terms of how females and males responded to ipratropium bromide treatment for chronic obstructive pulmonary disease, noting that improvements in forced expiratory volume were found in more females than males, and that the effectiveness of the treatment was impacted by BMI among females but not among males, indicating a sex-related responsiveness to pharmacological treatment [72].

There has been a push within the field of immunology for a precision vaccine approach that takes sex differences into account throughout the stages of vaccine development and delivery [73,74]. According to a 2013 editorial by Klein and Poland, vaccines may be formulated to be sex-specific, and sex may be considered when measuring vaccine response, adverse events, and the delivery of vaccines. In the same editorial, the authors highlight that AEs due to vaccines are more prevalent in women who have a stronger immune response to vaccines [73]. Moreover, the first International Precision Vaccines Conference in 2018 was dedicated to exploring the theory and process of precision vaccines and how to integrate sex and age factors in order to develop vaccines that are immunogenic in populations with varying degrees of immune response [74].

According to Kim et al. (2010), biological differences explain the distinctive experience of disease experienced by females and males. That is why these factors should be considered when developing and prescribing treatments for diseases although this is rarely done. To support this, these authors present findings that show that females wake faster than males after receiving anesthetics such as propofol and nitrous oxide and are more susceptible to experiencing side effects after receiving these compounds. They propose the following in order to address the sex bias in trials and treatment: (1) journals should call for the inclusion of sex differences in the design and analysis of findings, and label single sex studies accordingly and provide a rationale for their single-sex status, (2) regulatory and funding agencies should require appropriate inclusion of both males and females in clinical trials and also sex-specific analyses, (3) both clinicians and patients should be informed about sex-specific reactions to drugs, and (4) health organizations should encourage the participation of women in clinical trials [75]. Rodriquez, Aquino, and D’Ursi argue for systematic searches for sex related effects on PK and PD of drugs in the development process since there are sex differences in the therapeutic agents used to treat cardiovascular disease, lipid metabolism, and obesity. They underline the need to investigate the pathophysiology of sex hormones, sex specific genetic influences and interactions with environment to better understand the underlying sex differences involved in the PK and PD processes and mechanisms [76].

#### 3.5.3. Adverse Drug Reactions and Adverse Events (ADRs/AEs)

Several grey literature documents suggest that there are differences between the ADRs and AEs experienced by females and males and note the relevance of sex and gender-related factors for both ADRs and AEs. A 2011 editorial indicated that females and older individuals were more likely to develop myopathies when taking statins [77], while a 2010 editorial suggested that men may be more prone to hip fractures when consuming proton pump inhibitors (PPIs) due to the impact of this drug on calcium absorption [78]. The authors suggested that post-menopausal women taking calcium supplements could be sheltered from hip fractures caused by PPIs [78]. Data on ADRs from two internal medicine departments in Romania show that the majority of people affected were females. Using the Dose, Time and Susceptibility (DoTS) classification system, findings showed that 4% of all the reported ADRs was due to sex and these findings were in opposition to other research that used the same DoTS classification system without finding differences in sex-disaggregated susceptibility outcomes. For example, females had an increased risk of developing ACE inhibitor-related coughing and osteoporosis due to prednisone, whereas males had a higher tendency to develop gynaecomastia when using spironolactone and erectile dysfunction from metoprolol [79].

Drawing upon several antiepileptic drugs, a 2016 commentary discussed the influence of different sex and gender-related factors on efficacy and safety. The authors recommended the lowest possible dosage for women planning their pregnancy and that valproate should be used only if no other options are available. This is highly important for women since oral contraception, as well as physiological changes and poor adherence during pregnancy have effects on efficacy. Both men and women face safety issues due to hormone imbalances that can disrupt reproductive and sexual function. Furthermore, it was reported that valproate might be particularly risky for women due to its association with polycystic ovary syndrome, osteoporosis, and an increased risk of ADRs at the onset of menopause. It was also suggested that it may be linked to major congenital malformations and autism-spectrum disorders when taken while pregnant [80].

A 2010 opinion piece discussed the impact of different sex related factors such as body weight, drug exposure, hormones, etc. have on PK/PD mechanisms and ADRs. The authors highlighted the role of industry, regulators and academia and called for collaboration in order to better understand the role of the biological mechanisms involved in sex differences in ADRs [81].

#### 3.5.4. Policy Updates

Some articles discussed policy updates on HPV vaccination, QT interval prolongation studies and medications for treating insomnia. For example, a policy statement in 2012 suggesting updated recommendations for HPV vaccines considered sex and age in relation to who should be vaccinated, when, and with which HPV vaccine. This update was made in response to new safety and efficacy evidence of the available HPV vaccines and their ability to prevent cancers in both males and females associated with HPV and at what age they would be most effective [82].

According to Shah and Morganroth (2012), in the recommendations made by the ICH on the QT liability in drug development, there was a place for stronger recommendations to be made regarding the sex and sex-related factors. These statements were based on the evidence that there are differences in baseline QT intervals between men and women, as well as drug exposure sex differences in QT prolonging drugs, and evidence to support the potential for unknown sex differences of QT prolonging drugs beyond drug exposure, sensitivity, and body weight, such as the influence of sex hormones. These authors consider that it is not enough to conduct subgroup analyses by sex only when there is evidence to do so as per the ICH suggestion. This effort should be taken regardless of the evidence in order to be able to find differences between males and females [83].

Evidence from two articles in this group discussed different FDA warnings issued for zolpidem, an insomnia drug medication and the danger of impaired driving the following day [84,85]. When the FDA approved zolpidem in 1992, the label suggested a lower dosage for elderly individuals in comparison to younger adults. However, the label did not include any recommendations for females and males. These sex differences recommendations were included in 2011 due to the evidence on differences between females and males in the next-day blood drug levels. Research showed that 8 h after consuming zolpidem, 3% of men and 15% of women had concentrations of the drug in the blood that exceeded the cut off point for impaired driving. When looking into the extended-release form of zolpidem, it was found that 33% of women and 25% of men had concentrations of the drug in the blood that surpassed the impaired driving limit. This risk and the high reports of AEs led to the FDA issuing a safety notice, recommending lower zolpidem doses and requiring manufacturers to include different recommendations regarding the dosage by sex in 2013 [84,85].

## 4. Discussion

Drug policy and regulatory frameworks are determined by a mix of factors such as industry innovation, investment interests, market size, government policies, advocacy, scientific curiosity, regulatory mistakes, as well as consumer need and demand. This review describes varied sources of data from opinion pieces in grey literature, evidence from journal articles, advocacy based legacy websites and state sponsored regulatory documentation. Efforts to include women, females, sex, gender, and racial, ethnic and age groups in research, clinical trials and pharmacovigilance have steadily increased over the past 50 years, following certain catastrophic experiences with some drugs in the 1950s–70s that led to rigid exclusions of females, especially those of reproductive age. Notable among these were negative outcomes with thalidomide and diethylstilbestrol (DES) [3,4], drugs administered mainly during pregnancy and resulting in fetal deformities (thalidomide) and next generation female cancers (DES). In response, regulators and researchers excluded females and women from phase 1 clinical trials for several decades, citing reproductive potential, risk, and complexity [86].

Since then, advocates for women’s health and reproductive health, along with advocates for equity for race and ethnic groups have asked for more inclusion and specificity in drug testing and regulation and have made concrete suggestions for change. Both of the websites reviewed reflect long time advocacy efforts from inside and outside research communities, and address inclusion and the redress of exclusion by offering practical advice on trial design and engagement strategies. They both reflect the importance of overlapping EDI goals to the integration of SGBA+ in prescription drug management.

Currently, regulatory agencies are also increasingly acknowledging policies regarding EDI related roles and responsibilities along with increasing diversity and inclusion in clinical trials and research. They are beginning to hold their stakeholders and research partners accountable by prioritizing and facilitating the collection of quality data, applying SGBA+, fostering the inclusion of diverse populations, and requiring transparency of data. To achieve transparency, these efforts could be formalized by regulatory agencies in annual publications describing the demographics of clinical trial participants for new products, and progress toward EDI goals. However, as more research and government organizations impose EDI goals, it remains critically important to continue to maintain and develop a clear parallel SGBA+ integration in processes, in order to fully advance regulatory science [12], as EDI is in no way a replacement for SGBA+ in prescribed drug regulation or any other area.

Academics writing in opinion pieces or editorials have also offered numerous suggestions of how to address the integration of sex and gender related factors in PK/PD and AE, in clinical trials, and offered wide array of examples of conditions and drugs, illustrating the extent of this need. The advent and growth of sex and gender science has catalyzed and prioritized the relevance of measuring, analysing, and reporting upon sex and gender related variables in human health, and informing the regulation of prescription drugs is an important example of its utility. While not comprehensive, there is growing evidence on the impacts of sex and gender and PK/PD, and AE [17,24]. However, SGBA+ is rarely applied internationally in assessing such evidence and in formulating regulation and policy. And despite the presence of the ICH, there is little consistency across participating countries in the integration of SGBA+ issues in documentation.

Across the regulatory agencies that manage prescription drugs selected for this review, there is inconsistent and partial application of these concepts, data, and ideas. As described, the ICH has an important role in guiding numerous participating countries but does not illustrate or activate a consistent understanding of sex and gender conceptualization, language, science, or SGBA+. These omissions help perpetuate a lack of consistent integration of SGBA+ in drug management and miss providing leadership on these issues to participating countries. In contrast, the FDA has been a leader in acknowledging and addressing conceptual inaccuracies and omissions concerning sex and gender and rectifying its use of terms in its regulatory documents for managing prescription drugs. In addition, the FDA has made consistent and evolving overt attempts to include wider population representation by sex and race/ethnicity over the past 25 years. The EMA has also made efforts over the past 10 years to improve its guidance on trial inclusion, representation and reporting of efficacy with respect to sex, age, and ethnicity, but to our knowledge has not made an explicit commitment to an integrated SGBA+.

Like most of the regulatory documents we assessed, Health Canada documents were inconsistent in their inclusion of sex and gender and in their use of terminology. They did not consistently require data on sex categories (females or males) or gender (men or women or gender diverse people) across their documents. Guidance documents could mandate applicants and others for more precise data that reflect sex-related PK/PK factors and other biological concerns in keeping with a federal policy requiring GBA+ first initiated in 1999 and revised since, as well as with Canadian Institutes of Health Research (CIHR) guidance on SGBA+ [18] both of which are directly pertinent to reviewing prescription drugs. In addition, since its inception in 2019 the recommendations of the SAC-HPW have directly addressed these issues in managing prescription drugs in Canada, among other products utilized by women or those that have a sex and gender component. However, recent announcements indicate that changes are afoot. In October 2022, Health Canada initiated a process for requesting sex-disaggregated data from clinical trials regarding safety and efficacy to be included in new drug submissions [86]. Health Canada indicates that gender data are not being collected at this time, but that:

*“The questionnaire is the first step in a multi-phase approach to collecting disaggregated data from drug submissions. Other phases will include amending regulations and working internationally to promote the need for greater diversity when a drug is in development”* [84].

The language in all of these agencies’ documents could be more precise if ‘sex-related differences’ were replaced by ‘sex-related factors’ to account for dynamic and varied PK/PD processes that impact males and females differently or influence some groups of males or females in similar ways. Such a difference in terminology in regulatory documents would alert reviewers and applicants to the evidence in PK and PD studies that impacts all humans, while drawing attention to those factors that underscore or signal sex differences and/or similarities (e.g., hormones, anatomy, organ function, genetics, physiological processes, etc.).

## 5. Conclusions

The processes used to regulate prescription drugs result from and reflect contextual, scientific, political, precautionary, advocacy and market forces. Across the agencies reviewed, guidance documents, policies, and procedures do not routinely and consistently utilize sex and gender related terminology, or integrate PK/PD evidence into review processes, nor do they enable comparisons of AE data from diverse pharmacovigilance databases. Overall, many documents are sex and gender blind, misuse words and concepts, or are rooted in a ‘sex-difference’ paradigm that does not address sex- and gender-related processes and mechanisms of importance to assessing efficacy and safety of prescription drugs. Consequently, we do not know enough about risk, safety, or efficacy from a sex/gender perspective and how sex and/or gender interact with each other and with age and race/ethnicity to produce impacts and reflect or address health inequity. Even in the Canadian context with a well-established and mandated SGBA+ policy, sex and gender or diversity variables have not consistently integrated or reported upon in regulatory processes regarding prescription drug management.

There remains considerable room for improvement in ensuring safety and efficacy for both sexes and all genders, as well as age and race-based groups. First, an overarching commitment to improving and supporting sex and gender science would assist regulators in mandating requests for such data from applicants for new drugs. Second, recognizing the gaps in what we know and don’t know regarding sex, gender, age, and race/ethnicity and prescription drugs is essential in order to generate precision and precaution in regulations, product monographs and consumer information. Third, a clear commitment to data transparency and accountability regarding the PD/PK and AE issues affecting the diversity of the populations potentially consuming prescription drugs is required. Taken together, these three recommendations will ensure that the spirit and principles of SGBA+ EDI will be respected and enacted, and that both health and gender equity will be improved.

## 6. Limitations

This review has some limitations. We focused on exemplar publicly available guidance documents, operating procedures, standards advice, grey and academic literature from selected agencies identified in conjunction with our policy partners. Non-public regulatory and operational documents from these countries are not included. Therefore, our review may not be wholly reflective of all SGBA+ related policies or practices in the selected agencies. There may also be resources that integrate SGBA+ in the regulatory processes of prescription drugs that are in place in other agencies that we did not access in our review.

## Data Availability

Not applicable.

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
