# Peer review of "Sex, Gender, and the Regulation of Prescription Drugs: Omissions and Opportunities"

_ijerph, 2023, doi:10.3390/ijerph20042962_

Round 1

Reviewer 1 Report

Gender-related factors affect prescribing, access to drugs, and desire for specific prescribed therapies. This article draws on a policy-research partnership project that examined the lifecycle management of prescription drugs.

Abstract: Please clearly state your objectives in the abstract. the issue of clinical trials mentioned in your literature is not being reflected in the abstract. 

Methods: Please clearly state your inclusion and exclusion criteria. How the articles were selected? How the search strategy was established? 

 English language: There is a need to improve the flow of the manuscript. English language editing is required. 

Similarity index: The similarity index (turnitin report) is 49%. Please reduce it. 

Author Response

Thank you for your comments!

Reviewer 2 Report

The manuscript is very interesting. I would recommend accepting it in present form.

Author Response

Thank you for your review!

Reviewer 3 Report

Very interesting article and topic. Please find my comments below. 

1. Can you please provide some more information regarding sex and gender-based analysis plus and what this actually is? 

2. In your materials and methods you say that you reviewed "four documents from two websites" - what are these websites and how did you find them?

Author Response

Thank you for your comments and suggestions!
